

# Estimating the extent of Antarctic summer sea ice during the Heroic Age of Exploration

Tom Edinburgh[1*], Jonathan J. Day[1]

[1]Department of Meteorology, University of Reading, Reading, United Kingdom
[*]Currently: Department of Applied Mathematics and Theoretical Physics, University of Cambridge, United Kingdom

*Correspondence to*: J.J. Day (j.j.day@reading.ac.uk)

**Abstract.** In stark contrast to the sharp decline in Arctic sea ice, there has been a steady increase in ice extent around Antarctica during the last three decades, especially in the Weddell and Ross Seas. In general, climate models do not to capture this trend and a lack of information about sea ice coverage in the pre-satellite period limits our ability to quantify the sensitivity of sea

ice to climate change and robustly validate climate models. However, evidence of the presence and nature of sea ice was often recorded during early Antarctic exploration, though these sources have not previously been explored or exploited until now. We have analysed observations of the summer sea ice edge from the ship logbooks of explorers such as Robert Falcon Scott, Ernest Shackleton and their contemporaries during the Heroic Age of Exploration (1897-1917) and in this study, we compare these to satellite observations from the period 1989-2014, offering insight into the ice conditions of this period, from direct

observations, for the first time. This comparison shows that the summer sea ice edge was between 1.0° and 1.7° further north in the Weddell Sea during this period but that ice conditions were surprisingly comparable to the present day in other sectors.

## 1 Introduction

Understanding the interactions between polar sea ice trends and global climate change is of key importance in climate science. Amplified Arctic warming is closely associated with the dramatic reduction in Arctic summer sea ice, predominately through

ice-albedo feedback effects (Manabe and Stouffer, 1980; Serreze and Barry, 2011). However the reasons for the unexpected positive trend in Antarctic sea ice extent since the 1970s are not yet well understood (Parkinson and Cavalieri, 2012; Turner and Overland, 2009), and the climate models submitted to the Fifth Climate Model Inter-comparison Project do not reproduce this circumpolar satellite-era increase (Maksym et al., 2012; Turner et al., 2013). This has led the Intergovernmental Panel on Climate Change to assign *low confidence* to future projections of Antarctic sea ice extent in its latest assessment report (Stocker

et al., 2013).

This increase in pan-Antarctic sea ice extent is a sum of opposing regional trends, with large increases in the Indian Ocean, Weddell and in particular Ross Seas dominating over decreases in the Bellinghausen and Amundsen Sea (Parkinson and Cavalieri, 2012). Both mechanical and thermodynamic forcing by the atmosphere (Holland and Kwok, 2012) and changes to the Southern Ocean (Gille, 2002; Jacobs, 2002) are thought to play a role in these overall trends. It has been suggested that



anthropogenically driven changes such as ozone depletion (Ferreira et al., 2015; Sigmond and Fyfe, 2010; Turner et al., 2009), and responses to greenhouse gasses via negative sea-ice/ocean feedback (Zhang, 2007), and ice sheet runoff (Bintanja et al., 2013; Swart and Fyfe, 2013) all play a role. However, the relative importance of each of these processes is not well quantified (e.g. Turner et al., 2015). It is also likely that internal climate variability has played a part in the observed changes (Polvani and Smith, 2013), with a positive trend in the Southern Annular Mode thought to be a contributing factor (Lefebvre et al., 2004; Thompson and Solomon, 2002). Both a lack of sea ice data from the pre-satellite era and a lack of credibility in climate models restricts analysis of these processes (Abram et al., 2013). A longer term context may give some insight into our understanding of the dominant mechanisms.

Recent years have seen significant efforts in the recovery of historical meteorological records from ships logbooks (Brohan et al., 2009, 2010). Many of these logbooks contain detailed descriptions of the sea ice state at regular intervals and provide an invaluable source of sea ice edge information, but require careful interpretation (Ayre et al., 2015). Such data are available from the earliest Antarctic voyages in the 19th century - of Cook, Bellinghausen, Ross and others (Wilkinson, 2014) - but data from this early period are too temporally and spatially restricted for any firm conclusions to be made (Parkinson, 1990). It is not until the *Heroic Age of Exploration* that a sufficient level of data was collected to make concrete interpretations about the sea ice cover.

The period known as the *Heroic Age of Exploration* began with the Belgian Antarctic Expedition of 1897-99 and ended in 1917 with the conclusion of Shackleton's British Imperial Trans-Antarctic Expedition. This period saw an expansion of exploration around the continent (Fig. 1), allowing perhaps the earliest window for pan-Antarctic climate analysis using observed records. It is worth noting that the difficulty in travelling through heavy pack ice, much more of a problem during this period than in the current day, undoubtedly had a large influence on the expedition routes. For this reason, the sub-expedition voyages were undertaken mainly in the summer months (Nov-Mar), with some notable multi-year exceptions (such as the *Aurora* and *Endurance*, during the 1914-1917 Imperial Trans Antarctic Expedition), during which ships drifted, frozen into the pack ice, throughout the Antarctic winter.

Until now, evidence of Antarctic sea ice conditions during this period has only been available from proxy sources, such as those derived from ice cores. Chemical tracers within cores, such as sea salt and methane sulphonic acid (Curran et al., 2003), are known to co-vary with the latitude of the sea ice edge. These interactions are well understood in the Weddell Sea sector, where the century-long record of ice freeze and thaw dates at South Orkney Island (Murphy et al., 2014) has enabled such proxies to be well calibrated, showing a long term decline in both spring and autumn ice cover in this region (Abram et al., 2007). Nevertheless, it is currently unclear how generalizable these methods are to other regions of Antarctica, due to a limited set of directly observed sea ice data (Abram et al., 2013). The results of the present work will provide additional reference points for such studies.

20[th] century variations in sea ice cover have been inferred from whale catch positions, which provide a substantially more abundant, albeit less reliable and indirect, source of sea ice information (Vaughan, 2000). Previous studies, using these catch positions as a proxy record for the ice edge, suggested a 2.8° southward shift in the mean latitude of the summer sea ice edge,



equating to a 25% decline sea ice extent between 1931-1961 and 1971-1987 (Cotté and Guinet, 2007; de la Mare, 1997). However, there is some disagreement over the magnitude and spatial nature of results inferred from these records (Ackley et al., 2003; de la Mare, 2009), in part due to the accuracy of the satellite-derived ice during the summer melt season (Worby and Comiso, 2004) but also due to the evolution of whaling practices in the Southern Ocean (Vaughan, 2000). However, the ice

decline documented by de la Mare is consistent with a significant warming of the Southern Ocean between 1950-1978 documented recently by Fan et al. (2014). Whale catch records from earlier periods, in particular the *Heroic Age*, have not yet been used in sea ice analysis.

In this study, we use sea ice edge positions recorded in the ship logbooks during the *Heroic Age* to estimate the mean summer ice edge latitude, both regionally and for the whole Antarctic, during the period 1897-1917. We compare these with modern

satellite data in order to determine whether Antarctic summer sea ice extent was different to the present day and identify and quantify the possible changes.

## 2 Method

We use data collected during eleven expeditions of the *Heroic Age* for which frequent observations on the composition and nature of the sea ice were recorded (Table 1). Many of these logbooks had already been digitised recently as part of the

International Comprehensive Ocean-Atmosphere Data Set (ICOADS) initiative (Woodruff et al., 2011). Others needed to be digitised specifically for this study, either from photographic images of the original logbook or in person from the original logbook itself. These were combined to create a dataset of 191 observed ice edge positions (included in Supplementary Material) providing an almost circumpolar picture of the Antarctic summer sea ice edge (Fig. 2).

Logbooks from this period typically include meteorological observations taken at frequent intervals throughout each day.

Details of the sea ice cover were recorded along with a descriptive summary of the sea state and meteorology in the time period between the quantitative meteorological observations. Sometimes sea ice remarks were recorded in a specific column of the logbook describing the ice state, but more often included under a general observation heading that also encompassed additional comments regarding weather, sea state and other notable features, including wildlife. In some cases, the time of certain observations or events, such as when the ship passed from a region of consolidated ice cover into open water, were noted

specifically. These observations provide a clear picture of the presence and composition of the ice throughout the expeditions. We have only used expeditions for which a summary of the sea ice conditions was recorded at regular time intervals, either hourly or every four hours and have included a full list of terms used to describe the ice conditions in Table S1. We have assumed that the time the observations were recorded against reflects the time zone that the ship was operating in. Whilst this was certainly the practice in Royal Navy logbooks of the time, it may not be true for all non-British expeditions (Scott

Woodruff, personal communication). Logbook times were all converted into UTC prior to comparison with the PM satellite data. In most logbooks, the position of each ship was recorded only at midday. Geo-locating remarks about the ice at other times of day required linear interpolation between the midday positions of consecutive days based on the time associated with the ice information.



In order to identify the sea ice edge for the period, we used these remarks about the sea ice to estimate points in space and time where the ships were traversing or travelling along the sea ice edge. The aim, and therefore the route, of an expedition had a significant impact on the number of ice edge points that can be determined for each particular log and there is a clear distinction in the type of exploration (see Supplementary Videos).

Some expeditions, such as the *Terra Nova* expeditions to the Ross Sea, were primarily concerned with land exploration or the race to the South Pole and therefore the voyage of the ship was mostly a means of transport to the continent. In this scenario, it is common that the ship only crossed the ice edge on the journey south to and the return north from the continent; in which case, the position of the first and last observations of sea ice on the respective journeys are taken as the ice edge. However, the focus of other expeditions, such as the Australasian 1911-1914 expedition to the Western Pacific sector, on the *Aurora*, was

to explore undocumented coastlines by ship. Hence, the ship was often travelling along the sea ice edge for long periods, passing in and out of areas of sea ice frequently, and the ice edge is often unambiguously recorded in the logbooks using terms such as 'skirting pack ice'.

Interactions with the ice edge, in particular the navigation between open water and regions of ice cover, can sometimes be complicated by an ambiguity in the exact time that the ship encountered the ice edge. As an example, consecutive remarks in

the log of the Scotia expedition of 1903-04, made on the 1st March at 1100UTC and 16 hours later on the following day at 500UTC, observe 'steaming through loose pack' and 'no ice in sight' respectively. In cases such as this, this ambiguity makes it difficult to objectively identify the position of the ice edge, so we have taken the approach of trying to define a northern bound for the ice edge. We believe that a sensible approach is to utilise the entries recording the transition between these regions, marking as the ice edge the northernmost within such a pair of entries, thereby increasing our set of observations and

reducing the level of subjectivity in the inference of the ice edge from the terms used in the logs.

Before describing how we performed this comparison, it is useful to consider the relationship between these point data and satellite observed concentration. In satellite sea ice concentration products, a threshold of 15% is generally used to define the location of the ice edge. A detailed study by Worby and Comiso (2004) compared the sea ice edge as recorded by onboard trained human observers, during expeditions between 1989 and 2000 in the Western Pacific sector, with the 15% contour in

PM satellite imagery for the same day. They showed a high level of agreement during the Mar-Oct ice growth season but noted that during the summer melt season (Nov-Feb), the sea ice edge was systematically further north than the 15% contour in both the NASA TEAM (Cavalieri et al., 1984) and Bootstrap (Comiso, 1986) algorithms, with the Bootstrap algorithm being a closer match to the in-situ data. They argue that during this time of year, saturated bands of ice and floes, particularly at the edges of the pack ice, may be very localised, resulting in ice concentration below the 15% threshold when averaged over the

25km footprint of the PM instrument. In addition, these bands often comprise mostly brash ice, the PM signature of which can be almost indistinguishable from seawater. They estimate the offset between the observed ice edge and ice edge derived using the Bootstrap algorithm to be $0.75 \pm 0.61°$ during the melt season. However, it should be noted that their analysis has not been extended to other sectors, so we do not know how representative it is of the ice conditions outside of the Western Pacific. It is important to take this into account in the following analysis, since an apparent southward shift in the ice edge between the





*Heroic Age* points and the present day Bootstrap derived ice edge may overestimate the actual change, and should therefore best be considered as an upper bound.

As there is greater consistency with the in situ ice edge observations, we have chosen to use the Bootstrap algorithm daily sea ice concentration from the National Snow and Ice Data Centre (NSIDC) to estimate the present day sea ice edge, rather than NASA TEAM algorithm. From this, we calculated a daily mean sea ice concentration for the period 1989-2014, during which daily NSIDC data was available. Using this field, we have defined the daily satellite-observed mean ice edge as the contour joining the midpoints of the northernmost 25km pixels that do not exceed the 15% threshold in sea ice concentration. Then, for each ice edge observation from the logbooks, we computed the distance from the ship-observed position to each point on the contour for that particular calendar day using the Haversine formula and spherical law of cosines, and selected the latitude of the point on the contour with the minimum distance to the ship-observed position for the paired analysis. These differences were then averaged to calculate estimates for the mean change in the ice edge latitude for each sector and for the whole of Antarctica.

We believe the offset between the ice edge recorded by human observers and the satellite-derived ice edge to be the largest source of uncertainty in this analysis. Therefore, we use the Worby and Comiso value of 0.75° in the discussion of our results in order to address this source of uncertainty.

## 3 Results

The most dramatic change between the *Heroic Age* and the present day is in the Weddell Sea, where we have found that the ice edge was 1.71° further north during the *Heroic Age* and 0.96° further north with the inclusion of the Worby and Comiso offset, with both values significant at the 5% level (Table 2). This change agrees well with the observed decrease in land-fast ice at South Orkney Island during the last century (Murphy et al., 2014).

However, observations in the Weddell Sea are clustered into a small number of years (1903, 1904 and 1914), and as such may be influenced by natural year-to-year variability in the sea ice extent. Indeed, the ice edge observed by the crew of the *Scotia* in 1903 is particularly far north, even compared to observations from by the same crew the following year, and the greater ice extent in this region during the *Heroic Age* may be an exaggeration of the actual change between these two periods as a result. El Niño events often result in negative surface air temperature and sea ice anomalies in the Weddell Sea (Yuan, 2004) and it is possible that the large El Niño of 1902/03 may have had some influence in these anomalies. Nevertheless, our results indicate that many of the recorded ice edge positions lie further north than has been seen in any year since 1989 (Fig. 3, Supplementary Fig. 2, Supplementary Video 1).

The differences in other sectors appear to be much smaller. We observe statistically significant but small differences of 0.21° in latitude in the Bellinghausen and Amundsen Sea and 0.62° the Ross Sea, but no evidence of a significant difference in latitude in the Western Pacific and Indian Ocean region, which we have merged due to the limited data available in the Indian Ocean sector (Table 2, Fig. 3). However, as stated in the previous section, this is an upper bound for the observed change and



if we take into account the observed offset to the satellite data, the ice edge may actually have been further south in all of these areas (Table 2).

By averaging over all points, we also find the mean circumpolar change in the location of the ice edge. This pan-Antarctic shift of up to 0.41° southwards since the *Heroic Age* implies at most a 10.0% decrease in Antarctic sea ice extent between then and the present day (Fig. 4). This is much smaller than the 25% decrease between the 1950s and 1980s that was inferred by de la Mare (1997) from whaling records and suggests that, if we accept the results inferred from the whaling records, the sea ice was less extensive during the period 1897-1917) then it was during the period (1931-1961).

Although ice edge latitude is a convenient measure in which to analyse the data, it is more common in climate change assessment reports, such as the IPCC-AR5, to assess hemispheric sea ice variability and change using sea ice extent, rather than ice edge latitude (Vaughan et al., 2013). We do not have estimates of the ice edge position for all longitudes and therefore cannot calculate the ice extent during the *Heroic Age* from the logbook data alone. However, we were able to form estimates using the shape of the satellite sea ice climatology for each day within the DJFM period, by computing the number of 25km square pixels within region enclosed by the satellite-derived 15% contour and the corresponding contour whose coordinates have been shifted by the calculated change in latitude to give an approximate estimate of the ice edge during the *Heroic Age* for that day. We then averaged this over the DJFM period. This calculation also included regions enclosed by these two contours at polynya ice edges.

Our estimate of the mean DJFM sea ice extent, based on the mean ice edge latitude, is $10.45 \times 10^6 \text{km}^2$ (or $8.37 \times 10^6 \text{km}^2$, using the Worby and Comiso offset). Comparing our DJFM sea ice extent estimate to the Met Office Hadley Centre sea ice HadISST2.1 dataset (Titchner and Rayner, 2014), we find that our values are at least $1.59 \times 10^6 \text{km}^2$ lower (Fig. 4). During this period HadISST2.1 is based on a climatology for the period 1929-1939, derived from German Atlas charts (Deutsches Hydrographic Institute, 1950). Data from 185 whaling expeditions (mostly Norwegian but some from English whaling log books) were used as its basis (Titchner, H.; personal communication).

## 4 Summary and Discussion

In this study, we have used logbook data recorded explorers during the *Heroic Age of Exploration* to estimate an upper bound for the change between the summer sea ice edge during that period and the present day. To summarise our conclusions:

- We estimate that the DJFM sea ice edge was at most 0.41° further south between 1989-2014 than it was during the *Heroic Age* (1897-1917), implying a reduction of 10.0% in pan-Antarctic extent.
- This change is most dramatic and statistically robust in the Weddell Sea, where the ice edge has shifted 1.71° southward between the two periods.
- Our estimate of the change in extent between the *Heroic Age* and the present day is small relative to estimates of the change between the 1950s and 1970s, based on whale catch data (Cotté and Guinet, 2007; de la Mare, 1997; Titchner and Rayner, 2014). This suggests the possibility that the sea ice was significantly more extensive during the period 1931-1961 than during the *Heroic Ag*e.



Outside the Weddell Sea, the mean change in ice edge latitude is small compared to the known summer offset between in situ and satellite ice edges. It is therefore plausible that the sea ice extent in these regions has in fact experienced a small increase. Either way, the climate was much more similar to the present conditions than one might expect based on climate model simulations of the early 19th Century (e.g. Turner et al., 2013). These ice edge data, which we make available in the

supplementary material of this paper, could be used by climate model developers as a tuning target for pre-Industrial Antarctic sea ice cover.

It has been suggested that the use of whale catch positions for estimating the sea ice edge will overestimate any changes (Ackley et al., 2003); this may be responsible for the disparity between our results and those investigating the whaling data. However, it is also plausible that the Antarctic ice edge exhibits significant decadal and multi-decadal variability (Latif et al.,

2013), as has been observed in the Arctic (Day et al., 2012; Divine and Dick, 2006; Miles et al., 2014). If so, it is possible that there was an increase in sea ice extent between the *Heroic Age* and the 1930s, followed by the decrease between 1961 and 1971, suggested by de la Mare (1997). The MSA record from Law Dome, East Antarctica (Curran et al., 2003) and extensive sea ice in early 1960s Nimbus Satellite data (Gagné et al., 2015; Meier et al., 2013), seem to support this hypothesis.

We have excluded some of the expedition logbooks available to us because of the lack of ice condition observations, including

the *Southern Cross* expedition of 1898-1900, which recorded only an infrequent daily summary of the sea ice, and the *Hertha* and *Jason* expeditions of 1893-1894, neither of which recorded a summary of the sea ice. However, there are a few logbooks from this period, including those from the Norwegian expedition on the *Fram* (1910-1912), the Japanese expeditions on the *Kainan Maru* (1910-1912) and the German Expedition on the *Deutschland* (1911-1912), that have not yet been imaged and digitised and could potentially increase our knowledge of the sea ice conditions during this period. Similarly, although whaling

records for the *Heroic Age* are less complete than during the later period studied by de la Mare (1997, 2009), whaling records from the *Heroic Age* could provide additional validation of our findings.

Further analysis into the processes driving the long-term variations in Antarctic sea ice described in this study will be an important next step. In particular, expanding our source of Southern Ocean climatological data by increasing efforts to digitise ship logbooks from this data sparse region may be key to both our understanding recent Antarctic sea ice behaviour and

improving climate model performance.

**Acknowledgements**

Tom Edinburgh was funded by a NERC-SCENARIO funded studentship. Jonathan Day was funded by an AXA Post-Doctoral Research Fellowship and by the Walker Institute, University of Reading. We would like to thank Clive Wilkinson at the Climatic Research Unit, University of East Anglia, for his advice on logbook sources and for access to digitised logbook data.

Thanks also to the Royal Geographical Society Foyle Reading Room, for access to original logbooks from the Discovery 1901-04 and Terra Nova 1903-04 expeditions.





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

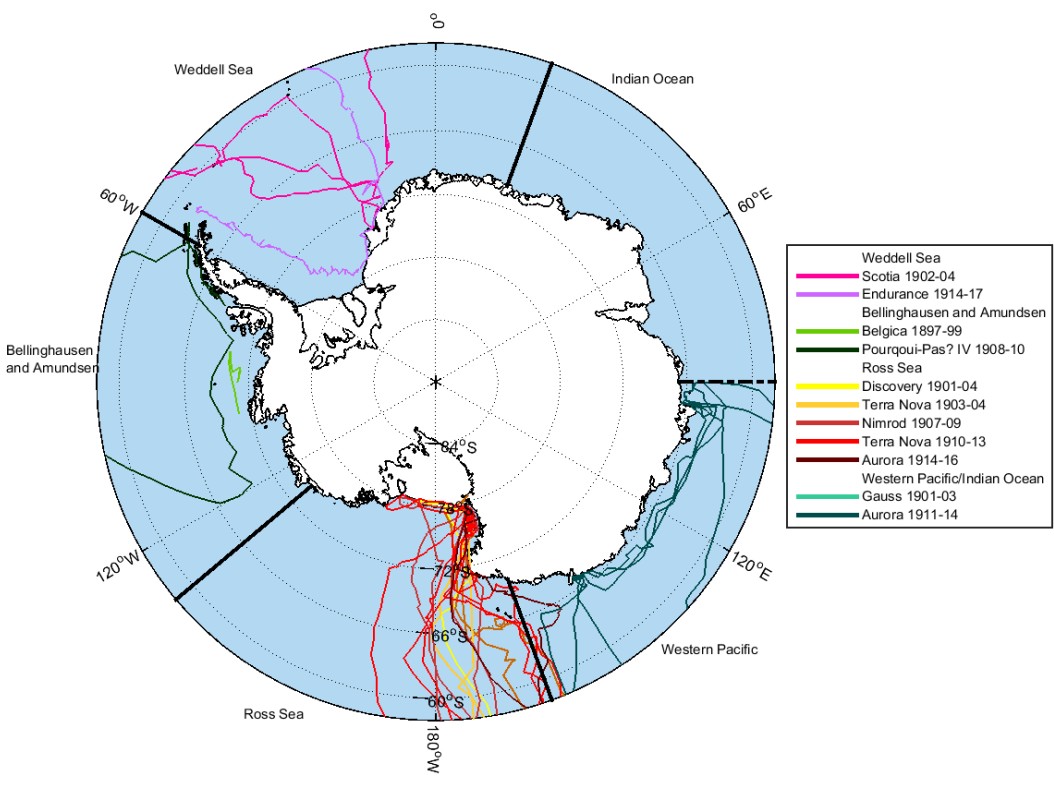

**Figure 1: Map of expedition routes taken by ships used in this study. We only have coordinates for entry and departure of the pack ice for the 1901-03 Gauss Expedition (Indian Ocean).**



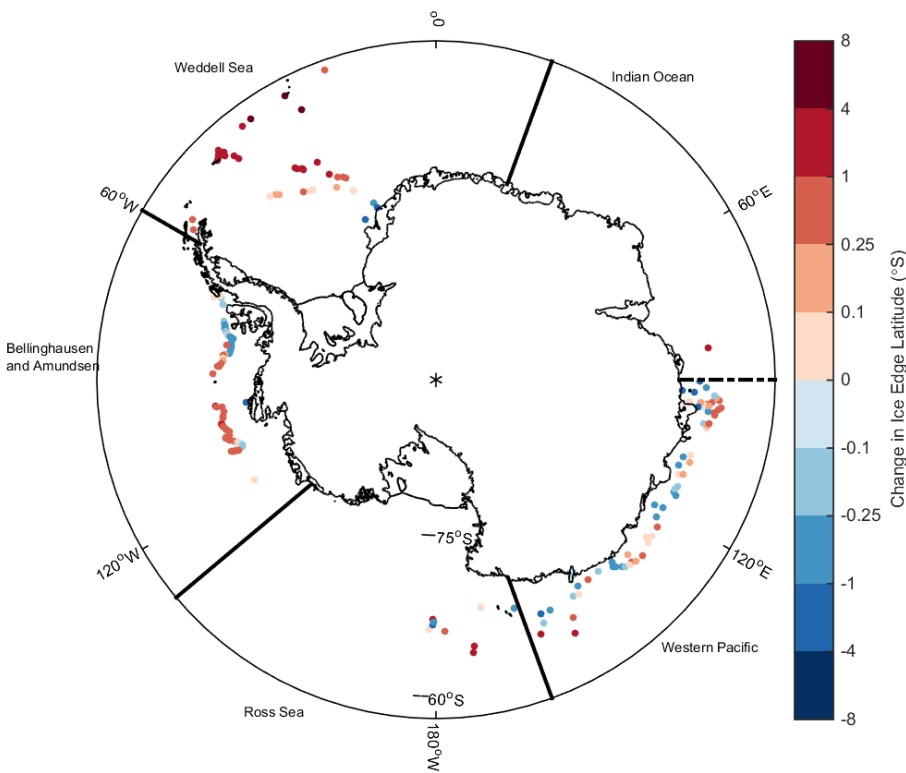

**Figure 2: Anomaly between ship-observed ice edge and the 1989-2014 mean PM-Bootstrap algorithm derived ice edge position for the appropriate calendar day. Anomalies are plotted at logbook position.**





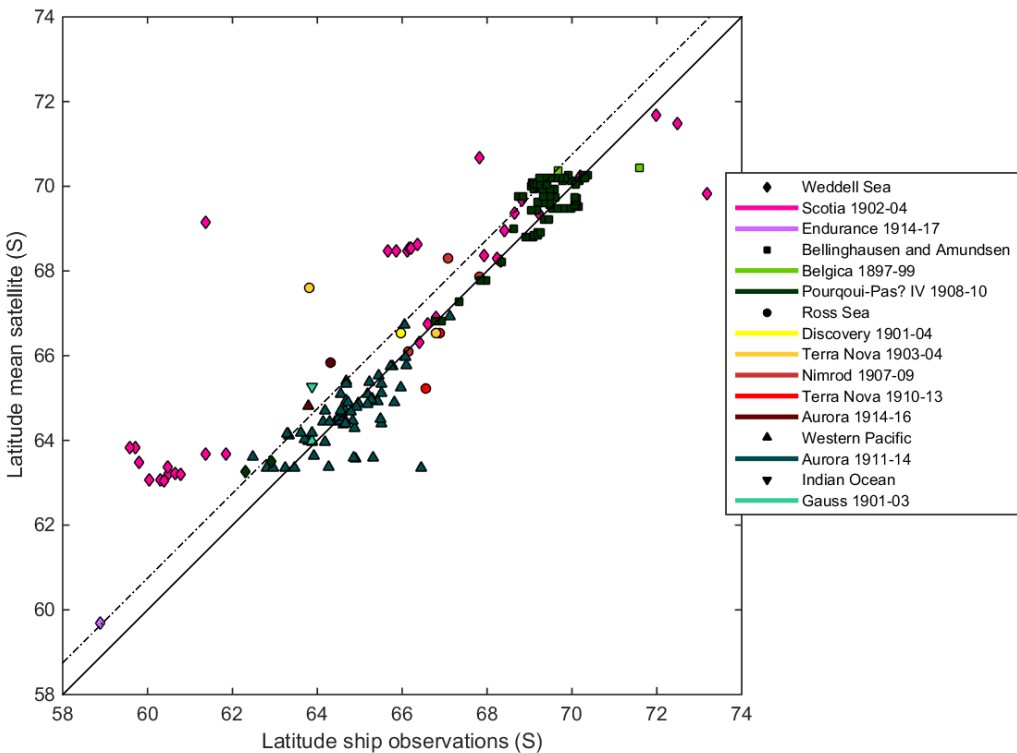

**Figure 3: Comparison of ship-observed and satellite-derived ice edge latitude, including a one-to-one line, which indicates no change in position. The dashed line provides an estimate of the southward offset one would expect when comparing the satellite-derived ice edge to in-situ ship observations, as calculated by Worby and Comiso (2004).**





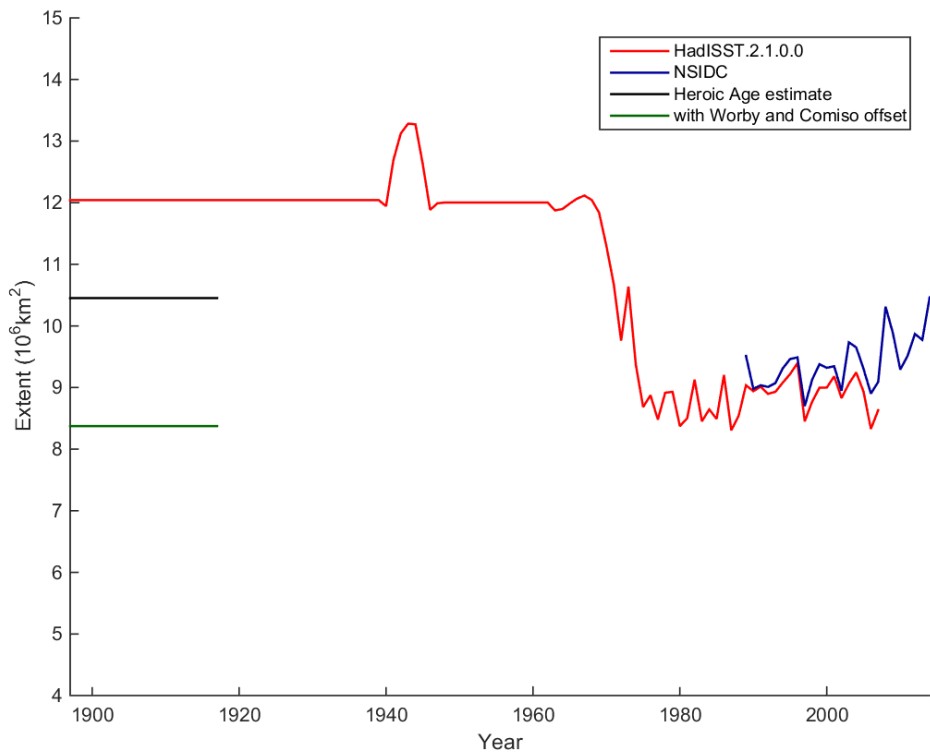

**Figure 4: The estimated DJFM Antarctic sea ice extent climatology for the period 1897-1917, with and without the inclusion of the Worby and Comiso offset, is plotted alongside time series of DJFM mean sea ice extent calculated from HadISST2.1 and NSIDC PM Bootstrap sea ice concentration datasets.**



| Date | Ship | Expedition | Expedition Leader | Sector | Source |
|---|---|---|---|---|---|
| 1897-99 | Belgica | Belgian Antarctic Expedition | Adrien de Gerlache | Bellinghausen and Amundsen | Arctowski (1904) |
| 1901-03 | Gauss | First German Antarctic Expedition | Erich von Drygalski | Indian Ocean | Arctowski (1904) |
| 1901-04 | Discovery | National Antarctic Expedition 1901 | Robert Falcon Scott | Ross Sea | Deck Logbook (Royal Geographical Society (AJP Collection) item no.: AA/21/1 to AA/21/6) |
| 1902-04 | Scotia | Scottish National Antarctic Expedition | William Speirs Bruce | Weddell Sea | ICOADS (Mossman, 1907) |
| 1903-04 | Terra Nova | National Antarctic Expedition 1901 | Robert Falcon Scott | Ross Sea | Deck Log book (Royal Geographical Society (AJP Collection)) |
| 1907-09 | Nimrod | British Antarctic Expedition 1907 | Ernest Shackleton | Ross Sea | ICOADS (Kidson, 1930) |
| 1908-10 | Pourquoi-Pas? IV | Fourth French Antarctic Expedition | Jean-Baptiste Charcot | Bellinghausen and Amundsen | Roach (1914) |
| 1910-13 | Terra Nova | British Antarctic Expedition 1910 | Robert Falcon Scott | Ross Sea | ICOADS |
| 1911-14 | Aurora | Australasian Antarctic Expedition | Douglas Mawson | Western Pacific | ICOADS (Mawson, 1939) |
| 1914-16 | Aurora | Imperial Trans-Antarctic Expedition | Aeneas Mackintosh | Ross Sea | ICOADS |
| 1914-17 | Endurance | Imperial Trans-Antarctic Expedition | Ernest Shackleton | Weddell Sea | ICOADS |

**Table 1: Expedition information and source materials used in this study. For items not digitised by ICOADS (Woodruff et al., 2011), the source material and archive are listed.**



|  | Pan-Antarctic | Weddell Sea | Bellinghausen and Amundsen Seas | Ross Sea | Western Pacific and Indian Ocean |
|---|---|---|---|---|---|
| Mean difference | 0.41 (-0.34) | 1.71 (0.96) | 0.21 (-0.54) | 0.62 (-0.13) | -0.08 (-0.83) |
| Variance of difference | 1.44 | 3.76 | 0.25 | 1.97 | 0.47 |
| Sample size | 191 | 35 | 80 | 10 | 66 |
| Wilcoxon SR test Approximate z value | -4.41 (6.21) | -4.17 (-2.85) | -3.31 (6.66) | -1.04 (0.74) | 0.46 (6.73) |
| p value Significance | <0.001 (<0.001) significant | <0.001 (0.022) significant | <0.001 (<0.001) significant | 0.099 (0.39) significant (not) | 0.33 (<0.001) not (significant) |

**Table 2: Mean differences in ice edge latitude between the ship-observed and daily mean satellite-derived ice edge position by Antarctic sector. Positive differences indicate where the ship-observed ice edge is north of the mean satellite-derived ice edge. Bracketed quantities refer to the same difference, but with the Worby and Comiso offset subtracted.**