# Peer review of "Estimating the extent of Antarctic summer sea ice during the Heroic Age of Exploration"

_The Cryosphere, 2016_

## Referee Comment (RC1) · D. Divine (Referee) · 16 May 2016

Review of a manuscript for *The Cryosphere*

**Estimating the extent of Antarctic summer sea ice during the Heroic Age of Exploration**

**Summary:**

In the manuscript the authors present a synthesis of sea ice observations from logbooks of eleven scientific expeditions to Antarctica during the so-called Heroic Age of Exploration in late 19$^{th}$-early 20$^{th}$ century. Due to apparent difficulties of navigation in sea ice covered waters for the vessels of that period the retrieved data is mostly represented by austral summer months. The observed local daily ice edge positions are compared with satellite observations of the modern era. The authors conclude that pan Antarctic summer sea ice extent must have exhibited little changes since the early 20$^{th}$ century, acknowledging nevertheless possible regional contrasts in the inferred centennial trends and a hidden imprint of the multidecadal variability on sea ice extent. A substantial discrepancy with the available sea ice climatology for the study period suggests that more efforts should be invested into recovery/processing the relevant historical data from various voyages to the region for the pre- IGY1957 period.

**Main comment**

The sparse and very fragmentary climate data for the period preceding the IGY, when the consistent exploration of the entire Antarctic region has begun, substantially limits our understanding of the present and future effects of the ongoing climate change in Antarctica. It concerns both the evaluation/improvement of the skills of GCMs and the interpretation of climate proxies from the region. In that respect the paper touches upon a very relevant and timely subject of exploration of the alternative sources of instrumental and observational data that can partly fill the existing data gaps. The Southern Ocean, at least during the austral summers, was a region of a noticeable international economical activity already since the end of the 19$^{th}$ century. Logbooks from numerous whaling and, occasionally, research vessels may therefore contain invaluable weather/sea ice information that yet to be uncovered. This paper represents one of the relatively few attempts made so far to systematize the sea ice observations from the early expeditions to Antarctica. The paper is generally well written and the results are presented consistently. I believe therefore that the manuscript certainly deserves to be published in *The Cryosphere* with relatively minor changes that are mainly related to the data presentation.

Please see my comments below:

Page 5 line 7: "…15% threshold in sea ice concentration…" Not entirely clear how the actual threshold concentration was estimated, is it based on a frequency of sea ice occurrence for a specific pixel or a mean concentration over the period of satellite observations for a specific day of year?
Page 6 line 14: … "corresponding contour whose coordinates have been shifted by the calculated change in latitude to give an approximate estimate of the ice edge during…"
Again, the procedure is somewhat unclear. Did the authors apply a radial scaling (with respect to latitude) fitting the modern pan-Antarctic SIE for a particular day of year to local sea ice observations available for that day? Such extrapolation should be supported by estimates, for example using the EOF analysis, demonstrating that most of the variance for this day (or period of the season) is represented by a single principal component.

Table 2: Line "Variance of difference"
I wonder how symmetric the derived distributions are and if using a normal pdf (and hence sample variance) is adequate?

---

## Referee Comment (RC2) · F. Fetterer (Referee) · 18 Jun 2016

(or Please see attached PDF.)

Review of Estimating the extent of Antarctic summer sea ice during the Heroic Age of Exploration, by Tom Edinburgh and Jonathan J. Day (doi:10.5194/tc-2016-90, 2016)

General comments:

The paper summarizes a carefully executed analysis comparing Antarctic ice extent from old data in the form of ship logbook entries with ice extent from today's satellite data. The authors acknowledge the obstacles inherent in comparing such different data types, and address them. Numerous references point to wide and deep research before the work with actual numbers began, and this adds weight to the results captured

in the paper.

The authors find that with the exception of extent in the Weddell Sea, ice extent now is not much different than it was in the Heroic Age (1897-1917). Between then and now, however, it may have been much more extensive, based on whaling records and the earliest Nimbus satellite imagery. This is an important finding because it suggests significant decadal and inter-decadal variability in southern hemisphere ice extent.

Publishing research results that come from old data like ship logbooks is important because it broadens recognition of international projects like ICOADS and Old Weather that are working to ensure that the observations are not lost.

Specific comments:

Page 4 lines 21-36: Fortunately, the Worby and Comiso study quantifies average differences in where a human observer would note the ice pack beginning on a voyage south, and where satellite data would put it. The Heroic Age did not have the "trained observers" of the Worby and Comiso study, and the study was only for one sector of the Antarctic, but given the other sources of error and imprecision when comparing satellite extent with that from logbooks, these are minor issues.

This section includes "They argue that during this time of year, saturated bands of ice and floes, particularly at the edges of the pack ice, may be very localised, resulting in ice concentration below the 15% threshold when averaged over the 25km footprint of the PM instrument. " The use of "footprint" is incorrect here. The algorithm uses brightness temperature from 37GHz, 22Gz, and 19GHz channels, and for these the field of view is larger. For the 19GHz frequency, it is about 70x45km. Simply replacing "footprint" with "grid cell size" works here though. (Substituting "grid cell" for "pixel" throughout would be more correct.) Page 5 lines 3 and 4: The actual data set the authors used needs to be cited properly. This is probably the right one to use: Comiso, J. C. 2000, updated 2015. Bootstrap Sea Ice Concentrations from Nimbus-7 SMMR and DMSP SSM/I-SSMIS, Version 2. [Indicate subset used]. Boulder, Colorado

USA. NASA National Snow and Ice Data Center Distributed Active Archive Center. doi: http://dx.doi.org/10.5067/J6JQLS9EJ5HU. [Date Accessed]. Page 5 lines 5 and 6: I'll just note that this way of constructing a mean ice edge for one day may not be optimal, because if the ice edge in some sector occupied a low latitude just a few times, and a high latitude most of the time, say, then the average concentration field from that day's 26 instances might easily have values >15% and place the average edge for that day in a place north of where it is likely to be seen. There are other ways to do it (e.g. the median edge used by the Sea Ice Index, described in https://nsidc.org/data/docs/noaa/g02135_seaice_index/#processing_overview), but for the purposes of this paper I don't think it makes much difference. Page 6 line 24: missing a "by" Supplementary material: The animations in the supplement do a great job of conveying information that the text covers but can't convey as well. Figure S2, Scatter Max is not the same as figure S2 within Supplementary Material (later lacks a trend line)

Please also note the supplement to this comment:
http://www.the-cryosphere-discuss.net/tc-2016-90/tc-2016-90-RC2-supplement.pdf

---

## Author Comment (AC1) · 11 Sep 2016

Response to reviewers: **Estimating the extent of Antarctic summer sea ice during the Heroic Age of Exploration**

*We would like to thank the reviewers for taking the time to carefully read and comment on this manuscript and for their encouraging and positive comments on the paper. In the following we outline our response to their specific comments and describe all changes. In addition to the suggestions by the reviewer we have also made two changes to the text based on comments received from Holly Titchener, to clarify the description of the methods in Section 2.*

**Reviewer 1:**

Page 5 line 7: "…15% threshold in sea ice concentration…" Not entirely clear how the actual threshold concentration was estimated, is it based on a frequency of sea ice occurrence for a specific pixel or a mean concentration over the period of satellite observations for a specific day of year?

*It is the latter, so 15% concentration in the mean sea ice concentration field for a given day of the year averaged over the period 1989-2014, which we have calculated for all days of the year. We have adjusted the text to make this clearer.*

Page 6 line 14: … "corresponding contour whose coordinates have been shifted by the calculated change in latitude to give an approximate estimate of the ice edge during…" Again, the procedure is somewhat unclear. Did the authors apply a radial scaling (with respect to latitude) fitting the modern pan-Antarctic SIE for a particular day of year to local sea ice observations available for that day? Such extrapolation should be supported by estimates, for example using the EOF analysis, demonstrating that most of the variance for this day (or period of the season) is represented by a single principal component.

*We did indeed apply a simple radial scaling method to estimate a pan-Antarctic mean extent for the period. We have adjusted the text to make the method used clearer. We are not asserting that all variability lies in a single principal component but there is simply not enough data to do anything more sophisticated in terms of estimating extent. The procedure and the scarcity of data used to approximate this means that there is an imprecision to our estimates that certainly warrants further investigation. We fully acknowledge these limitations and discuss these in the revised text. However, we do believe that putting our findings, regarding latitudinal variations of the ice edge, into the context of other estimates of areal extent during this period, such as HadISST, is important for the interpretation of these results.*

Table 2: Line "Variance of difference" I wonder how symmetric the derived distributions are and if using a normal pdf (and hence sample variance) is adequate?

*The distributions are not normal, which is why we used a non-parametric statistical test. However, we state the variance here as a useful measure for the reader to gauge the variability.*

**Reviewer 2:**

Page 4 lines 21-36: Fortunately, the Worby and Comiso study quantifies average differences in where a human observer would note the ice pack beginning on a voyage south, and where satellite data would put it. The Heroic Age did not have the "trained observers" of the Worby and Comiso study, and the study was only for one sector of the Antarctic, but given the other sources of error and imprecision when comparing satellite extent with that from logbooks, these are minor issues.

*This is a good point, which we now mention at Page 6, line 20 of the revised manuscript.*

This section includes "They argue that during this time of year, saturated bands of ice and floes, particularly at the edges of the pack ice, may be very localised, resulting in ice concentration below the 15% threshold when averaged over the 25km footprint of the PM instrument. "
The use of "footprint" is incorrect here. The algorithm uses brightness temperature from 37GHz, 22Gz, and 19GHz channels, and for these the field of view is larger. For the 19GHz frequency, it is about 70x45km. Simply replacing "footprint" with "grid cell size" works here though. (Substituting "grid cell" for "pixel" throughout would
be more correct.)

*We have replaced 'footprint' with 'pixel' as suggested.*

Page 5 lines 3 and 4: The actual data set the authors used needs to be cited properly. This is probably the right one to use:
Comiso, J. C. 2000, updated 2015. Bootstrap Sea Ice Concentrations from Nimbus-7 SMMR and DMSP SSM/I-SSMIS, Version 2. [Indicate subset used]. Boulder, Colorado USA. NASA National Snow and Ice Data Center Distributed Active Archive Center. doi: http://dx.doi.org/10.5067/J6JQLS9EJ5HU. [Date Accessed].

*We used the Bootstrap algorithm distributed as part of the NSIDC merged dataset. We have added the correct dataset citation.*

Page 5 lines 5 and 6: I'll just note that this way of constructing a mean ice edge for one day may not be optimal, because if the ice edge in some sector occupied a low latitude just a few times, and a high latitude most of the time, say, then the average concentration field from that day's 26 instances might easily have values >15% and place the average edge for that day in a place north of where it is likely to be seen. There are other ways to do it (e.g. the median edge used by the Sea Ice Index, described in https://nsidc.org/data/docs/noaa/g02135_seaice_index/#processing_overview), but for the purposes of this paper I don't think it makes much difference.

*We acknowledge this comment by the reviewer. However, because we agree with their assessment that it is unlikely to affect the results, we do not modify the analysis.*

Page 6 line 24: missing a "by"

*Added as suggested:*

Supplementary material:

Figure S2, Scatter Max is not the same as figure S2 within Supplementary Material (later lacks a trend line)

*This figure has been replaced.*

*In addition to changes made in response to reviewers comments, we have made a number of changes to the manuscript following feedback from Holly Titchener, developer of HadISST at the Met Office:*

*Page 4 line 24: We used ship logbooks with data recorded at frequent time intervals and not fixed regular time intervals.*

*Page 5 line 25: Clarification about the ice edge points used depending on the direction of travel of the ship.*

*Figure 4: We have updated our HadISST2.1.0.0 curve to the latest version, HadISST2.2.0.0. We also found that the NSIDC Bootstrap and Heroic Age estimates had been plotted incorrectly and have corrected this. We have also added the NASA Team algorithm sea ice extent.*